# MAVS is important for antiviral defense against influenza A virus in a human respiratory epithelium model

**Maja Hemberg**[ID]**, Anne Louise Hansen, Jacob Storgaard, Julia Blay-Cadanet, Alice Pedersen, Anne Laugaard Thielke, Christian Kanstrup Holm**[ID]*

Department of Biomedicine, Aarhus University, Aarhus, Denmark

* holm@biomed.au.dk

## Abstract

The respiratory epithelium is an important immunological barrier and the first line of defense against influenza A virus (IAV). In mice and in various cellular systems, induction of type I interferons (IFNα/β) during IAV infections is known to depend on cytosolic RNA sensors retinoic acid-induced gene I (RIG-I) and melanoma differentiation-association gene 5 (MDA5) and their common adaptor protein mitochondrial antiviral-signaling adaptor protein (MAVS). Until now, it has not been possible to directly assess the importance of MAVS for induction of IFNs and for resistance to IAV infection in primary human respiratory epithelium. Here, we used CRISPR-Cas9 to establish MAVS-deficient cultures of primary human respiratory epithelium using the air-liquid interphase culture system. Using this setup, we show that MAVS is indeed required for the induction of type I and type III IFNs and subsequently for the induction of IFN-stimulated genes in response to IAV infection in this respiratory epithelium model. Finally, we demonstrate that MAVS is important for restricting viral replication in this model. In conclusion, this study demonstrates that MAVS plays a non-redundant protective role during IAV infection in primary human respiratory epithelium.

## Introduction

The respiratory epithelium is the primary site of influenza A virus (IAV) infection and is a key initiator of antiviral defenses [1]. IAV is an important airborne virus, causing pandemics and seasonal endemics with great societal impact [2,3]. Vaccines and antivirals against IAV exist, however, vaccines must be renewed annually due to viral mutations, and antivirals are only effective if administered very early [2,4,5]. These challenges highlight the importance of understanding innate immunity, especially at the viral entry point, since this might provide insights to improve the development of more effective antiviral therapies.

Upon IAV infection, viral RNA is recognized by RIG-I-like receptors, including the retinoic acid-induced gene I (RIG-I) and melanoma differentiation-association gene

**Data availability statement:** All relevant data are within the manuscript and its Supporting information files.

**Funding:** Novo Nordisk Fonden (NNF):Maja Hemberg,Anne Louise Hansen,Jacob Storgaard,Julia Blay-Cadanet,Alice Pedersen,Anne Laugaard Thielke,Christian K Holm 0066798; Frimodt-Heineke Fonden (Frimodt Heineke Fonden) The funders had no role in study design, data collection and analysis, decision to publish, or preparation of the manuscript.

**Competing interests:** The authors have declared that no competing interests exist.

5 (MDA5) [6–9]. Both signal through the mitochondrial antiviral-signaling adaptor protein (MAVS, also known as ISP1, VISA or CARDIF [10–13]). Activation of MAVS leads to phosphorylation of the transcription factors interferon regulatory factors 3 and 7 (IRF3/7) and nuclear transcription factor-κB (NF-κB) that drive the production of interferons (IFNs) and subsequentially the induction of interferon stimulated genes (ISGs), which are essential for viral restriction [6,10]. MAVS is antagonized by IAV and other RNA viruses, emphasizing its importance in antiviral responses [14]. MAVS-deficiency is various experiments have consistently demonstrated that MAVS is required for the production of type I IFNs (IFN-α and -β) during viral infections in cell types and mice [10,15–19]. However, discrepancies regarding the extent to which MAVS contributes to antiviral defense are still present. Some studies show increased viral replication and higher susceptibility to RNA viruses in the absence of MAVS [10,15,17], whereas others find no effect of MAVS KO on viral titers or survival following IAV infection [18,20]. Only a few studies have examined the importance of MAVS in the antiviral defense against IAV, and these studies report no consistent phenotype [9,16,18,20]. Thus, the potential non-redundant protective role of MAVS during IAV infection remains unresolved. In particular, the function of MAVS within the respiratory epithelium, the first line of defense against IAV, has not been directly investigated in a physiologically relevant human model. Understanding epithelial-intrinsic MAVS signaling is important for determining how early innate immune responses contribute the outcome of IAV infection.

In this study, we demonstrate that a functional knockout of MAVS in a physiologically relevant model of human respiratory epithelium decreased production of type I and III IFNs and ISGs following IAV infection. Furthermore, we show that MAVS deficiency increases viral replication. These findings indicate that MAVS is important for the interferon response and protection against IAV infection in this human respiratory epithelium model.

## Results

### Functional KO of MAVS by CRISPR-Cas9 in HAE-ALI cultures

To investigate the importance of MAVS in respiratory epithelium we utilized a model where primary human airway epithelium cells (HAE) was cultured in the air-liquid interface (ALI) system. This model contains ciliated cells and mucus producing secretory cells which are hallmarks of respiratory epithelium, thus mimicking the biology of the human airway [21]. The model was established by collecting respiratory epithelial cells from the nasal cavity of donors and dedifferentiating them into basal cells. Genetic modification using CRISPR-Cas9 was performed to generate MAVS knockout cells or AAVS1 knockout cells as a control. The KO cells were plated on Transwell membranes under liquid-liquid conditions. Once fully confluent, the apical medium was removed, initiating differentiation into a respiratory epithelium (Fig 1A), containing ciliated and secretory cells (Fig 1B). The KO was validated by Western blot, showing a decreased expression of MAVS in the MAVS KO culture compared to the AAVS1 KO culture (Fig 1C). To validate disruption of MAVS-dependent signaling, HAE-ALI cultures were infected with Sendai virus (SeV). SeV is a well characterized

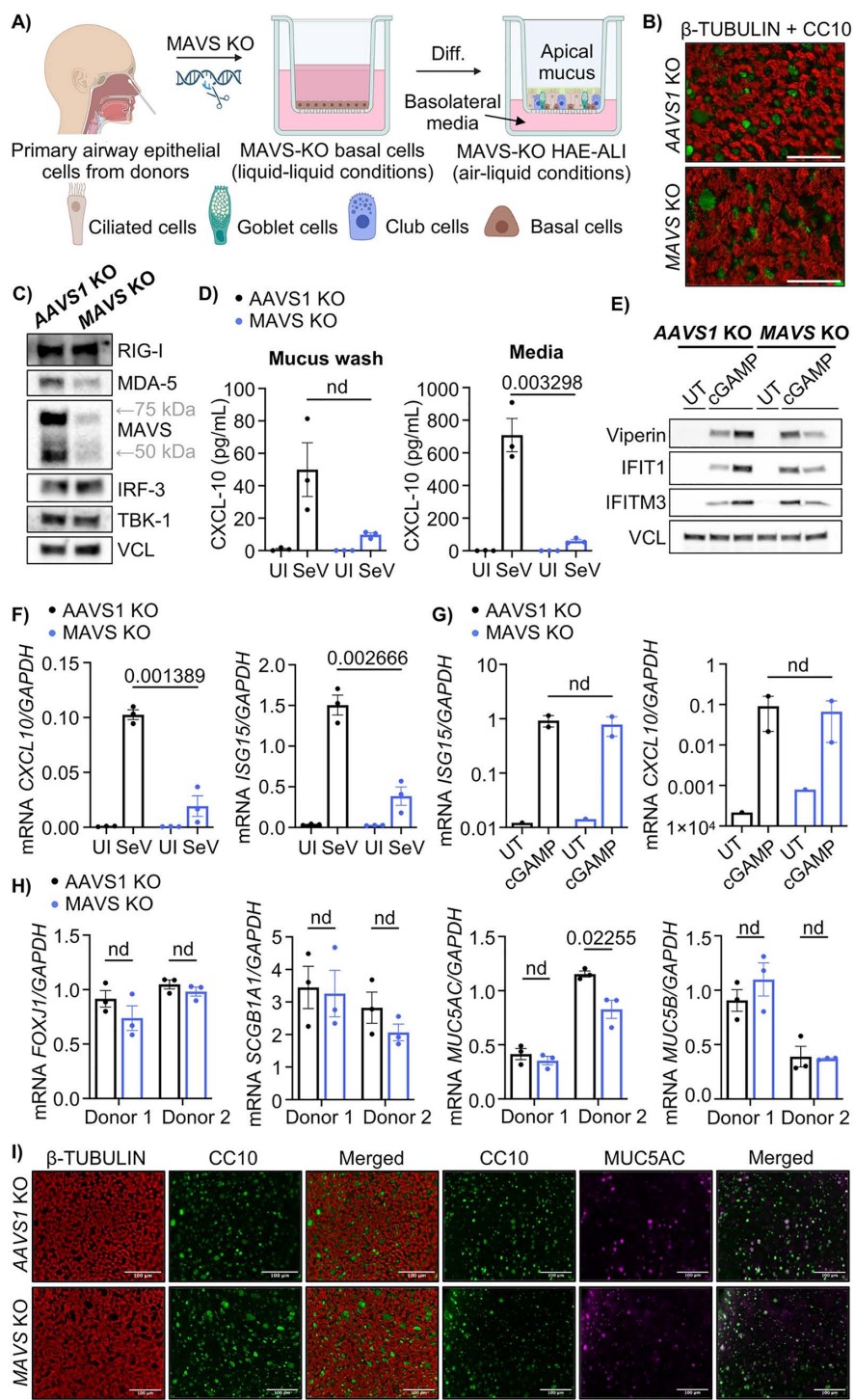

**Fig 1. Functional knockout of MAVS by CRISPR-Cas9 in HAE-ALI cultures. (A)** Schematic of the human airway epithelium (HAE) cultured in the air-liquid interface (ALI) system. Epithelial cells were isolated from the nasal cavity of healthy donors. CRISPR-Cas9-mediated knockout of AAVS1 or MAVS was introduced in basal cells, which were subsequently seeded onto transwell membranes and differentiated into respiratory epithelium under ALI conditions. **(B)** Immunofluorescence images of uninfected HAE-ALI showing cellular markers for ciliated cell in red (β-TUBULIN) and club cells in green (CC10). Scale bar = 50 μm. **(C)** Western blotting of AAVS1 or MAVS KO HAE-ALI monolayer cultures using specific antibodies for the MAVS-pathway. Vinculin (VCL) was used as a loading control. **(D)** AAVS1 KO and MAVS KO HAE-ALI cultures were infected with SeV (30 HAU/mL) or left uninfected for

16 hours. A hCXCL-10 ELISA was performed on the apical mucus wash and basolateral media. Each data point represent an independent culture (n = 3) derived from the same donor. **(E)** AAVS1 or MAVS KO HAE-ALI cultures were stimulated with 2'3'-cGAMP (6 μg/mL) or left untreated for 16 hours. The ISG levels were analyzed using Western blotting with VCL as loading control and with cell lysate from independent cultures in each lane, all derived from the same donor. **(F)** RT-qPCR analyzing ISG levels in SeV infected (30 HAU/mL) or uninfected HAE-ALI 16 hours post-infection. Data points represent independent cultures (n = 3) from the same donor. Normalized expression $2^{-\Delta CT}$. **(G)** AAVS1 or MAVS KO HAE-ALI cultures stimulated with 2'3'-cGAMP (6 μg/mL) or left untreated for 16 hours. ISG levels were analyzed using RT-qPCR. Each data point represents a normalized expression $2^{-\Delta CT}$ of independent culture (untreated n = 1, treated n = 2). **(H)** RT-qPCR was performed on uninfected AAVS1 or MAVS KO HAE-ALI cultures derived from two independent donors, analysing markers of ciliated cells (*FOXJ1*), club cells (*SCGB1A1*) or goblet cells (*MUC5AC* and *MUC5B*). Each data point represents one independent donor-derived culture (n = 3). Normalized expression $2^{-\Delta CT}$. **(I)** Immunofluorescence images of uninfected HAE-ALI showing cellular markers for ciliated cell in red (β-TUBULIN), club cells in green (CC10) or goblet cells in magenta (MUC5AC). Representative images are shown. Scale bar = 100 μm. For all panels, bars represent mean ± s.e.m. Statistical differences were determined by multiple unpaired t-tests with FDR correction using the two-stage step-up method of Benjamini, Krieger and Yekuteili (desired FDR = 5%). Q-values are shown above each comparison; "nd" indicates q-value > 0,05. Statistical analysis was not performed for untreated or uninfected groups (D, F, **G**).

activator of the RIG-I pathway and is widely used as a positive control in studies of antiviral innate immunity [8,10]. Infection with SeV induces a strong and reproducible activation of downstream signaling, thereby allowing validation of pathway integrity. Infection with SeV increased production of C-X-C Motif Chemokine Ligand 10 (CXCL-10) and Interferon Stimulated Gene 15 (ISG-15), measured by ELISA and qPCR. However, this induction was decreased in the MAVS KO cultures compared to the control (Fig 1D, F). Next, the specificity of the KO was tested by stimulating the cultures with the STING ligand cGAMP, which is known to induce IFNs independently of the MAVS-pathway [22]. This resulted in similar expression levels of Viperin, Interferon-Induced Protein with Tetratricopeptide Repeats 1 (IFIT1) and Interferon-Induced Transmembrane Protein 3 (IFITM3), as measured by Western blot (Fig 1E), as well as similar induction of CXCL-10 and ISG-15, as measured by qPCR, between the KO cultures (Fig 1G). This demonstrated that the KO of MAVS was specific and selectively affected the RIG-I and MDA-5 pathways in the HAE-ALI cultures. Next, we examined whether MAVS KO affected the cellular composition of the cultures. HAE-ALI cultures from two independent donors (donor 1 and donor 2) were assessed by qPCR for expression of Forkhead Box J1 (FOXJ1), Secretoglobin Family 1A Member 1 (SCGB1A1, also known as CC10), Mucin 5 AC (MUC5AC) and Mucin 5B (MUC5B). FOXJ1 was used as a marker for ciliated cells, SCGB1A1 (CC10) as a marker of club cells, and MUC5AC and MUC5B as markers of goblet cells [23]. No differences were observed in the levels of ciliated or club cell markers between the KO cultures, but the level of MUC5AC, a goblet cell marker, was lower in the MAVS KO culture from donor 2 compared to the AAVS1 KO. However, the level of MUC5B in this donor was similar between KO cultures, suggesting that the MUC5AC reduction was donor-specific and not a general effect of MAVS KO (Fig 1H). To investigate cellular morphology, whole HAE-ALI membranes were fixed and stained with antibodies against protein markers of the various cell types. The membranes were imaged using fluorescence microscopy. No differences in morphology or composition were observed between AAVS1 or MAVS KO HAE-ALI cultures (Fig 1I). These data suggest that MAVS-deficiency induced by CRISPR-Cas9 does not affect cellular composition, but does reduce the ISG response to SeV infection, indication that MAVS is important for the immune response in this model of human respiratory epithelium.

## MAVS is essential for the interferon response in HAE-ALI after IAV infection

To test whether the absence of MAVS is important for the interferon response in respiratory epithelium during IAV infection, we infected the HAE-ALI cultures with IAV (A/PR/8/34, H1N1) for 16 hours or 48 hours. At 16 hours post-infection the induction of type I (IFN-β) and type III interferons (IFN-λ), measured by qPCR, was decreased in the MAVS KO cultures compared to the control (Fig 2A). The same reduction was observed in the MAVS KO cultures at 48 hours post-infection and these findings were consistent across two separate donors (Fig 2B). Furthermore, qPCR analysis of CXCL10 and ISG15 showed that both genes were significantly reduced in the MAVS KO cultures compared to the control at 16 hours (Fig 2C) and 48 hours (Fig 2D) post-infection. When infecting with either IAV or SeV, Western blot analysis showed an

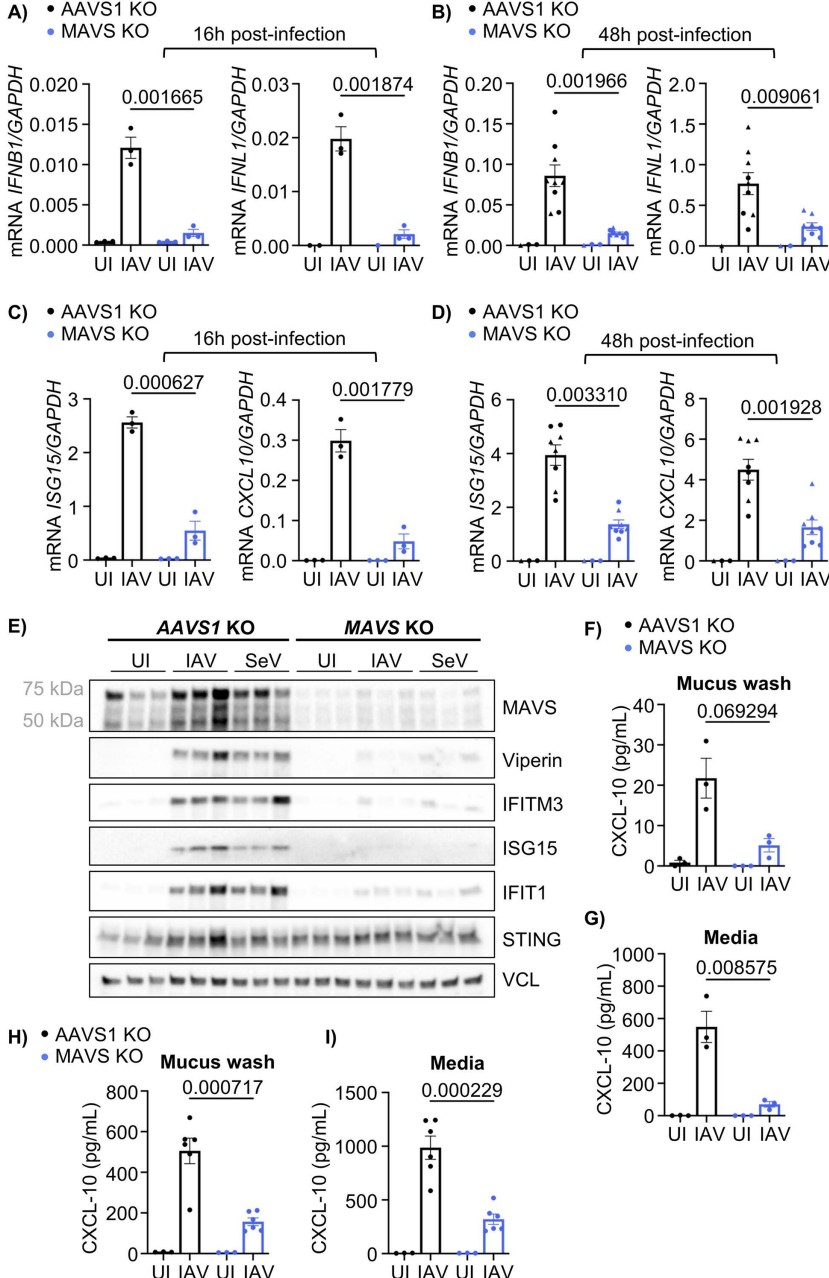

**Fig 2. MAVS is essential for the interferon response in HAE-ALI after IAV infection. (A)** RT-qPCR analysis of *IFNB1* and *IFNL1* in AAVS1 or MAVS KO HAE-ALI cultures infected with IAV (MOI 0.5) or left uninfected for 16 hours. Each data point represents an independent culture (n = 3) derived from the same donor (donor 2). Data is representative of two independent experiments (Fig A in S1 Fig). For *IFNL1*, some uninfected cultures had undetectable levels. **(B)** RT-qPCR analysis of *IFNB1* and *IFNL1* in AAVS1 or MAVS KO HAE-ALI cultures infected with IAV (MOI 0.5) or left uninfected for 48 hours. Pooled data from two independent experiments (donor 1 = ● (uninfected n = 2, infected n = 4), donor 2 = ▲ (uninfected n = 1, infected n = 5)). Each data point represents an independent culture. Statistical significance was determined by multiple paired t-test (FDR = 5%). **(C)** RT-qPCR analysis on ISGs in cultures infected with IAV or left uninfected (like in A) for 16 hours. Each data point represents an independent culture (n = 3) from the same donor (donor 2). Data is representative of two independent experiments (Fig B in S1 Fig). **(D)** RT-qPCR analysis of ISGs in AAVS1 or MAVS KO HAE-ALI cultures infected with IAV (MOI 0.5) or left uninfected for 48 hours. Pooled data from two independent experiments (donor 1 = ● (uninfected n = 2, infected n = 4), donor 2 = ▲ (uninfected n = 1, infected n = 4)). Each data point represents an independent culture. Statistical significance was determined by multiple paired t-test (FDR = 5%). **(E)** Western blot analyzing protein expression of MAVS, various ISGs and STING in AAVS1 or MAVS KO HAE-ALI cultures infected with either IAV (MOI 0.5) or SeV (30 HAU/mL) or left uninfected for 16 hours. Vinculin (VCL) was used as a loading control. Each lane

represents an independent culture (n = 3) derived from the same donor. **(F)** A hCXCL-10 ELISA was performed on cell-free apical mucus wash from AAVS1 or MAVS KO HAE-ALI cultures infected with IAV (MOI 0.5) or left uninfected for 16 hours. Each point represents an independent culture (n = 3) from the same donor. **(G)** Basolateral media from the same experiment as (F) analyzed by hCXCL-10 ELISA (n = 3). (H) hCXCL-10 ELISA on cell-free apical mucus wash from AAVS1 or MAVS KO HAE-ALI cultures infected with IAV (MOI 0.5) or left uninfected for 48 hours. Each data point represents an independent culture (uninfected n = 3, infected n = 6) from the same donor. Data is representative of two independent experiments (Fig D in S1 Fig). (I) hCXCL-10 ELISA on basolateral media from the same experiment as **(H)**. Each data point represents an independent culture (uninfected n = 3, infected n = 6) derived from the same donor. Data is representative of two independent experiments (Fig E in S1 Fig). For all panels, bars represent mean ± s.e.m. Unless otherwise stated, statistical differences were determined by multiple unpaired t-tests with FDR correction using the two-stage step-up method of Benjamini, Krieger and Yekuteili (desired FDR = 5%). Q-values are shown above each comparison; "nd" indicates q-value > 0,05. Statistical analysis was not performed for uninfected groups (A-D and F-I).

induction of Viperin, IFITM3, ISG15 and IFIT1 in the AAVS1 KO cultures following 16 hours of infection. This induction was decreased in the MAVS KO cultures alongside the MAVS protein levels, but the induction of STING was similar between KO cultures (Fig 2E). Reduced protein expression of ISGs in MAVS KO cultures was also observed at 48 hours post IAV-infection, although this reduction was less pronounced (Fig C in S1 Fig). Next, the release of CXCL10 from the HAE-cultures was examined by ELISA. This revealed a decreased amount of CXCL10 in the apical mucus wash as well as the basolateral media from the MAVS KO cultures compared to the control at both 16 hours (Fig 2F-G) and at 48 hours post-IAV infection (Fig 2H-I). Collectively, the lack of MAVS impaired the induction of both IFNs and ISGs, demonstrating that the MAVS pathway is important for initiating the immune response against IAV in this model of respiratory epithelial cells.

### MAVS is non-redundant for the protection against IAV in HAE-ALI

Considering that MAVS was important for the IFN response in the HAE-ALI, we wanted to investigate if the reduced MAVS expression also affected the viral replication in these cultures. To examine this, viral RNA corresponding to the Nucleoprotein (NP), Matrix Protein 2 (M2), and Non-structural Protein 1 (NS1) of IAV from infected AAVS1 or MAVS KO HAE-ALI was quantified by qPCR. At 48 hours post-IAV infection, viral RNA levels were increased in the MAVS KO cultures compared to the control for all viral genes. This finding was consistent across two independent donors (Fig 3A). This effect of MAVS KO was observed only after longer infections, as viral gene levels were similar between the two KO cultures at 16 hours post-infection (Fig F, G in S1 Fig). The expression of the viral NS1-protein was analyzed by Western blot, showing an increase of NS1 in the MAVS KO culture compared to the control (Fig 3C). This finding was consistent across two separate experiments and quantification showed a significant increase in the NS1 expression normalized to the loading control (Fig 3B). Next, we examined entire HAE-ALI membranes by immunofluorescence. Infected or uninfected AAVS1 or MAVS KO cultures were fixed 16 hours after infection or mock infection. The membranes were stained with DAPI and an antibody against the IAV protein NS1 and imaged using a fluorescence microscope. Quantification of NS1-positive cells showed that approximately 10% of cells were NS1-positive in AAVS1 KO cultures, whereas MAVS KO cultures exhibited a higher mean of approximately 13% NS1-positive cells (Fig 3D-E). Based on the immunofluorescence images, it was not possible to determine if any specific cell type was more likely to be infected, however, viral protein was observed in cells with a ciliated appearance in both cultures (Fig 3F). Together, increased viral RNA levels, higher accumulation of viral protein by Western blot, and a greater proportion of NS1-positive cells indicate that the viral replication of IAV is enhanced in the MAVS-deficient respiratory epithelium model.

## Discussion

Together, our results demonstrate the establishment of a physiologically relevant MAVS-deficient model of the human respiratory epithelium. The findings build on prior studies identifying MAVS as a central mediator of antiviral signaling; however, its specific role in the respiratory epithelium has not previously been investigated. Using a primary human airway

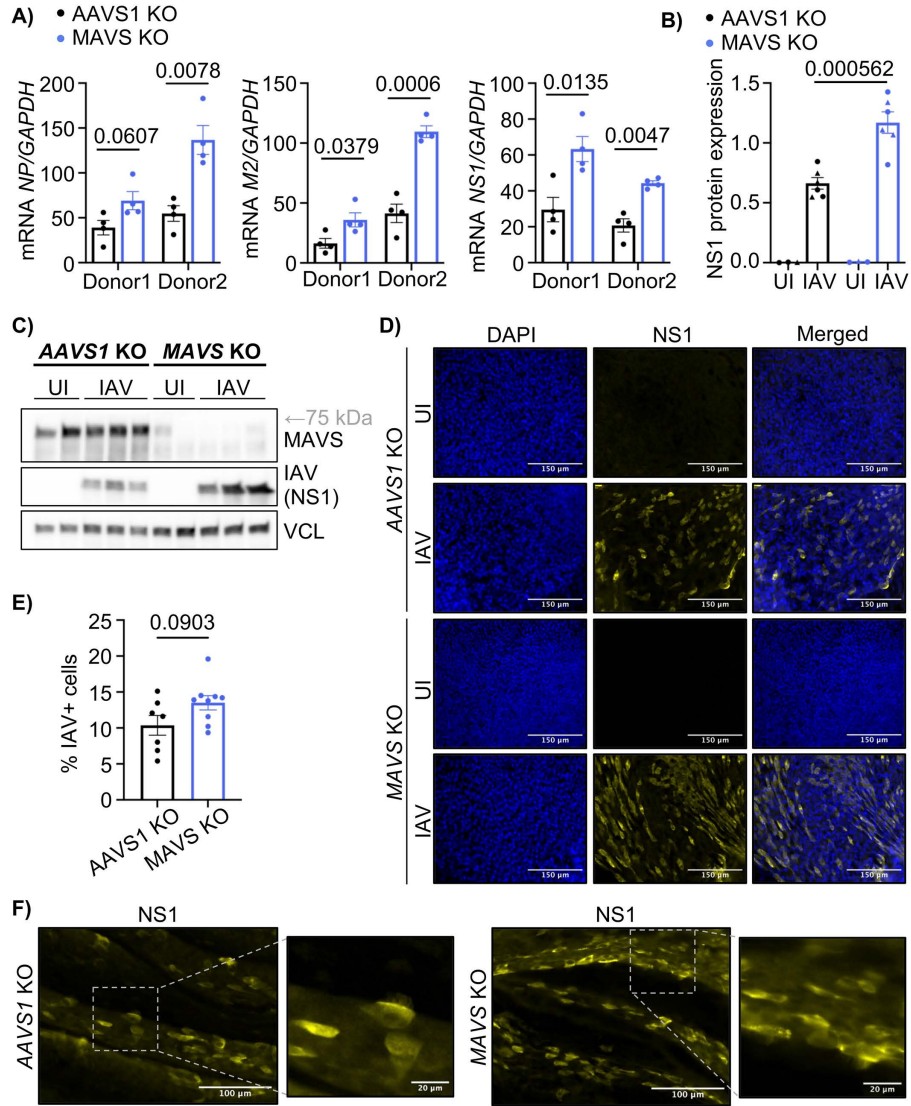

**Fig 3. MAVS is non-redundant for the protection against IAV in HAE-ALI. (A)** RT-qPCR analysis of viral RNA from three different IAV segments in AAVS1 or MAVS KO HAE-ALI cultures infected with IAV (MOI 0.5) for 48 hours. Each data point represents an independent culture (n = 4) derived from two different donors. Statistical significance was determined using a Welch's t-test and p-values are shown above comparisons. **(B)** Quantification of NS1 protein levels determined by Western blotting and normalized to the loading control (VCL). Pooled data from two independent experiments (donor 1 = ● (uninfected n = 2, infected n = 3), donor 2 = ▲ (uninfected n = 1, infected n = 3)). Each data point represents an independent culture. Statistical significance was determined using a multiple unpaired t-test (FDR = 5%). **(C)** Western blot analysis of viral (NS1) and MAVS protein levels in AAVS1 or MAVS KO HAE-ALI cultures infected with IAV (MOI 0.5) or left uninfected for 48 hours. Vinculin (VCL) was used as a loading control. Each lane represents an independent culture (uninfected n = 2, infected n = 3) derived from the same donor. Data is representative of two independent experiments (Fig C in S1 Fig). **(D)** Immunofluorescence images acquired at 40x magnification of AAVS1 or MAVS KO HAE-ALI cultures derived from the same donor infected with IAV or left uninfected for 16 hours. Nuclei were stained with DAPI (blue) and viral NS1 protein with an anti-NS1 antibody (yellow). Representative images of each condition are shown. Scale bar = 150 μm. **(E)** Quantification of immunofluorescence (IF) images of AAVS1 or MAVS KO HAE-ALI cultures 16 h post-infection with IAV (MOI 0.5). The graph shows the proportion of NS1 + cells relative to the total cell count. Cell counting was performed in ImageJ. Each data point represents one 40x IF image (AAVS1 KO, n = 12; MAVS KO, n = 13) from a single culture per condition. Statistical difference was determined using a Welch's t-test, and the p-value is shown above the comparison. **(F)** IF images acquired at 40x magnification of AAVS1 or MAVS KO HAE-ALI cultures 16 h post-infection with IAV. Images are derived from the same donor. A magnified inset of boxed of regions are shown to the right. Scale bars 100 μm (main) and 20 μm (insets). Bars indicate mean ± s.e.m. FDR correction by the two-stage step-up method of Benjamini, Krieger and Yekuteili (desired FDR = 5%). The q-values are indicated above comparisons for **(A-B)**. Statistical analysis was not performed for uninfected groups **(B)**.

epithelial (HAE-ALI) model, we provide direct evidence of epithelial-intrinsic antiviral responses regulated by MAVS. We show that MAVS is required for the induction IFNs and ISGs following IAV infection, and that viral replication is increased during prolonged infection in its absence in this HAE-ALI model.

Consistent with our findings, most studies examining the role of MAVS during RNA virus infections show decreased induction of type I and type III interferons following MAVS KO or silencing [10,15–19]. Many studies also report an effect of MAVS deficiency on the production of ISGs. Regarding the antiviral properties of MAVS, some studies indicate that it is important for protection against various RNA viruses [10,15,17]. The importance of MAVS in IAV infections has only been investigated in a few studies using MAVS-deficient mice. While these studies show that MAVS is essential for the induction of IFNs and ISGs upon IAV infection, supporting the findings of the current study, this did not translate into increased antiviral protection in the mice [18,20]. These studies were conducted in MAVS KO mice, so it remains unclear whether the findings apply to primary human cells. Moreover, they do not allow conclusions about the isolated role of MAVS in the respiratory epithelium in protection against IAV.

To our knowledge, this is the first study to investigate MAVS function in a primary human respiratory epithelial model. Our data suggest that, in the absence of MAVS, sensing of viral RNA by RIG-I or MDA5 fails to trigger the induction of IFNs and ISGs. This induction appears to be important for restricting viral replication in the respiratory epithelium during prolonged IAV infections. Although increased viral transcripts and altered viral protein levels in MAVS KO cultures indicate enhanced viral burden, infectious viral titers were not directly quantified by plaque-forming or TCID-50 assays, which remain the gold standards for assessing replication kinetics. Therefore, while our data support increased viral activity in the absence of MAVS, the extent to which this reflects an increased production of infectious virions cannot be definitively determined and needs further investigation.

Although MAVS protein levels were markedly reduced, low residual expression remained detectable by western blot. HAE-ALI cultures are heterogeneous, and electroporation may not affect all cells, making complete MAVS elimination challenging. Residual IFNs and ISGs induction in MAVS KO cultures may reflect incomplete gene disruption in a subset of cells and/or compensatory signaling through alternative pathways.

Although both ciliated and secretory epithelial populations were confirmed in KO and control cultures, gene editing efficiency was not assessed at single-cell resolution. Given that IAV exhibits strain- and cell-type-dependent tropism, heterogeneous editing across epithelial subtypes may influence observed antiviral responses. However, MAVS KO cultures from both donors consistently showed reduced protein expression compared to controls, supporting effective pathway disruption at population level and allowing investigation of the role of MAVS in this respiratory epithelium model. Future studies employing single-cell transcriptomic approaches and higher-resolution imaging with co-staining of epithelial subtypes will help define MAVS disruption across epithelial populations and provide insight into strain-dependent tropism and epithelial integrity following infection.

Nevertheless, analysis of FOXJ1, SCGB1A1, MUC5AC, and MUC5B expression, together with β-tubulin staining, indicated no overall changes in epithelial composition. However, we cannot exclude subtle effects of gene editing on epithelial differentiation or composition, which might influence specific cell populations.

This study is limited by the use of two same-sex donors. Although responses were consistent across both donors, inclusion of additional donors of both sexes would improve generalizability. In addition, the HAE-ALI model lacks immune cell components, and therefore does not capture epithelial-immune interactions that contribute to antiviral defense in vivo. As such, our findings reflect epithelial-intrinsic MAVS function rather than whole-tissue immunity.

The use of the laboratory-adapted influenza A/PR/8/34 (H1N1) strain represents another limitation. While this strain was selected due to its widespread use and ability to provide a robust and reproducible infection model, it has undergone extensive laboratory adaptation. Consequently, the host-pathogen interactions observed in this study may differ from those elicited by contemporary clinical influenza isolates. This represents a limitation of the present study, and future work using circulating human influenza strains will be important to determine broader applicability. Notably, inclusion of SeV, a robust activator of

RIG-I signaling, supports that the observed MAVS-dependent effects are not restricted to IAV but reflect a more general role in antiviral protection of the respiratory epithelium. Despite these limitations, the use of primary human respiratory epithelium cells remains a major strength of this study, providing a physiologically relevant system to investigate antiviral signaling.

In vivo, antiviral immunity arises from complex interactions between epithelial, innate immune, and adaptive immune cells, making it difficult to isolate tissue-specific contributions. The HAE-ALI model used in this study enables focused investigation of epithelial-intrinsic antiviral responses during the early phase of infection, prior to substantial immune cell recruitment. Therefore, this model provides unique insight into the first line of antiviral defense at the respiratory barrier. Furthermore, this model based on primary human cells could potentially limit the use of infectious animal studies.

Improving our understanding of epithelial innate immunity at the site of viral entry is important for the development of antiviral strategies against IAV. This study demonstrates that MAVS plays a non-redundant role in regulating interferon responses in human respiratory epithelium infected with IAV. However, these findings should be interpreted within an epithelial-only system, as complex interactions between epithelial and immune cells in vivo contribute to antiviral defenses. Therefore, the extent to which MAVS-dependent signaling is compensated by other pathways in more complex systems remains to be determined. Overall, MAVS KO significantly impairs IFN and ISG induction in this physiologically relevant human respiratory epithelial model, highlighting a critical role for MAVS in respiratory epithelial antiviral defense.

## Methods and materials

### Establishment of the air-liquid interface epithelium model

Primary nasal epithelial cells were isolated from healthy donors using an interdental brush (Brage Nilsson, P-My x-small). The brush was inserted in an angle following the bridge of the nose and twisted at the nasal turbinates. Cells were rinsed off the brush by gently expelling Monolayer medium (Airway Epithelial Cell Basal Medium (PromoCell, #C-21260) + 1 pack of Airway Epithelial Cell Growth Medium Supplement (PromoCell, #C-39160) + 100 U/mL Penicillin/Streptomycin (Gibco, #15140122)) and PBS (Biowest, #L0615). The cells were spun down at 130 x g for 5 min and resuspended in 2 mL Monolayer medium before being moved into a T10 flask (Thermo Scientific, #156367) precoated 24 hours earlier with 0,1 mg/mL Bovine type I collagen solution (Sigma-Aldrich, #804592, diluted in sterile ddH$_2$O). Once the monolayer cells were approx. 80% confluent, they were split into a T75 flask (Sarstedt, #83.8311) pre-coated with collagen using 1x Trypsin with 0,3 mM EDTA (10x Trypsin, (Gibco, 2,5%, #15090), diluted to working conc. in PBS + UltraPure 0,5 mM EDTA (Invitrogen, #15575)). When the cells were confluent in the T75 flasks, they were ready for genetic alteration. The collection and culturing of primary cells were approved by the Danish Research Ethics Committees (Case nr.: 1-10-72-182-19). Both donors in this study were male.

### CRISPR-Cas9 KO

Monolayer cells were genetically modified twice to ensure efficient MAVS knockout, using two sequential rounds of CRISPR-Cas9-mediated knockout of either MAVS or AAVS1 (control). Both knockout rounds followed the same procedure; however, after the first KO, the cells were seeded into T75 flasks pre-coated with 0,1 mg/mL Bovine type I collagen solution and maintained in Monolayer medium until they had recovered and reached sufficient confluency for the second KO. Following the second KO, cells were seeded onto 6,5 mm Transwell membranes (Corning, #3470) pre-coated with 30 µg/mL Bovine type I collagen solution (diluted in sterile ddH2O) with 75,000 cells pr. membrane.

The monolayer cells were detached from their flasks using Trypsin/EDTA. Trypsin was inactivated by DMEM (low glucose, Gibco, #11880-028) with 5% FBS. Cells were spun down at 130 x g for 5 minutes, resuspended in Monolayer medium, and counted using a Bürker-Türk chamber. Genetic modification was done by CRISPR-Cas9 KO of MAVS and AAVS1 (control). The sgRNAs used for the experiments were purchased from Synthego:

AAVS1 sgRNA: 5'-GGGGCCACUAGGGACAGGAU-3'

MAVS sgRNA #1: 5'- UGUCUUCCAGGAUCGACUGC-3'

MAVS sgRNA #2: 5'- CCGGUUCCCUGAGAGUGUGC-3'

The sgRNAs were resuspended and their concentrations were adjusted to 3200 ng/µL using TE-buffer (provided with sgR-NAs). RNP complexes for each sgRNA were prepared in PCR tubes by mixing 0.6 µL of Alt-R S.p. Cas9 nuclease V3 (IDT, # 1081059) with 3,2 µg of sgRNA pr. electroporation (max. 200.000 cells pr. electroporation). After 15 minutes at room temperature, the RNP complexes for MAVS #1 and MAVS #2 were mixed. For the initial KO, the entire cell suspension was used, however, for the second KO, a volume of the cell suspension corresponding to 75.000 cells pr. membrane was used. The cells were spun at 130 x g for 5 minutes, resuspended with PBS and spun down once more. After removing the supernatant, the cells were resuspended in 20 µL Opti-MEM (Thermo Fisher, #31985062) pr. electroporation. The cell suspension in Opti-MEM was divided between the two PCR tubes containing the AAVS1 or MAVS RNP-complexes. The solutions were added to an electroporation strip (Lonza, Nucleocuvette Strips, #V4XC-2032) and electroporated using the Lonza 4D-Nucleofector with the program DC.100 and the cell type P3. After electroporation, the cells rested for 2 minutes. Monolayer medium or Submerged medium (DMEM + 200 U/mL Pen/Strep mixed 1:1 with 2x ALI medium (Airway Epithelial Cell Basal Medium (PromoCell, #C-21260) + 2 packs of Airway Epithelial Cell Growth Medium Supplement (PromoCell, #C-39160) but without triiodothy-ronine + 1 mL of 1.5 mg/mL BSA) was used to remove the cells from the electroporation strip. The cell suspension in Mono-layer medium (1st KO) was added two separate T75 flasks. The cell suspension from the 2nd KO was seeded and submerged with Submerged medium on Transwell membranes. When cultures reached full confluency, ALI (Ali-Liquid Interface) was intro-duced and ALI medium (Pneumacult ALI medium kit (StemCell, #05002) + ALI medium supplement (StemCell, #05003) + 100 U/mL Pen/Strep, supplemented with 24 µg of hydrocortisone (StemCell, #07925) and 0.2 mg heparin (StemCell, #07980)) was added to basolateral compartment. The cells differentiated for at least 21 days, verified by ciliated beating and mucus produc-tion. Cells were kept at 37°C, 5% $CO_2$. Throughout the establishment and KO of the model, media was changed three times a week. Once movement of the cilia was observed, the cultured were washed weekly with DMEM to remove excess mucus.

### Infection or treatment of HAE-ALI

Before treatment or infection, the HAE-ALI cultures were washed for 5 minutes with DMEM to remove mucus build-up, and the ALI medium in the basolateral compartment was changed. For cGAMP treatment, 150 µl of DMEM or a dilution of 6 µg/mL 2'3'-cGAMP (InvivoGen, #tlrl-nacga23-02) in DMEM was added to the apical surface. After 1 hour in the incubator at 37°C, the apical solutions were removed, and the cultures were placed in the incubator for 16 hours. Infection of the HAE-ALI was performed by adding a virus dilution in DMEM, or by mock-infecting the cultures with DMEM, for 1 hour at 37°C before removing the solutions a returning the cultures in the incubator for 16 or 48 hours.

At time of harvest, the HAE-ALI cultures were washed for 5 minutes with DMEM and the basolateral media was removed, saving both the wash-solutions and media for ELISA. 400 µL of Trypsin/EDTA was added basolaterally and 150 µL was added to the apical surface. After approximately 5 minutes, the cells were resuspended in the Trypsin/EDTA solution and transferred to a tube with 500 µL of DMEM with 5% FBS. Cells that remained attached to the membrane were gently scraped off with the pipette tip. Cells were pelleted at 300 x g for 5 minutes, resuspended in cold PBS, and divided into two tubes. The tubes were centrifuged again at 300 x g for 5 minutes, after which the cells were lysed for either RNA isolation or Western blot.

### Viruses

For the infection studies, we used Sendai virus, Cantell strain (#10100816, Charles River) with 6000 HAU/ml, which was diluted 1:200 in DMEM (30 HAU/ml). Furthermore, we used the Influenza A/PR/8/34 (H1N1) strain (#10100374, Charles River) with an MOI of 0.5.

## Westen blotting

We used Western Blotting to detect proteins in our samples. Protein was isolated from the samples using a lysis buffer consisting of ice-cold RIPA lysis buffer (Thermo Fisher Scientific, #89901) supplemented with 5 IU/mL Benzonase (Sigma-Aldrich, #E1014), 10 mM Sodium Fluoride, and 1x complete protease inhibitor cocktail (Roche, #5892988001). Protein concentrations were determined using a BCA protein assay kit (Thermo, #23225). Before Western Blotting, the sample was mixed with XT Sample Buffer (Bio-Rad, #1610791) and XT Reducing Agent (Bio-Rad, #161-0792) and denatured at 95°C for 2–3 minutes. The amount of protein loaded in each well was approximately 7,5 µg and the Precision Plus Protein Dual Color Standard (Bio-Rad, #161-0374) was used as a protein ladder. Using a 4–20% Criterion TGX Precast Midi Protein Gel (Bio-Rad, #5671094 or #5671095), samples was separated by SDS-Page running at 70V for approx. 15 min followed by approx. 50–70 min at 120V. The protein was transferred to a PVDF membrane (Bio-Rad, #170-4157) using a Trans-Blot Turbo Transfer from Bio-Rad. The membranes were blocked for 1 hour at room temperature using PBS/Tween (PBS, Biowest, #X0515, added to 4,5 L of deionized $H_2O$) with 0.05% Tween 20 (Sigma-Aldrich, P1379) (PBS/Tween) mixed with 5% skim-milk powder (Sigma Aldrich, #70166). After blocking, the membranes were rinsed with PBS/Tween and cut to match the molecular weight range of the protein of interest. The membrane pieces were incubated ON at 4°C with the following antibody solutions diluted in PBS/Tween: anti-RIGI (Cell Signaling, monoclonal, rabbit, #3743, 1:1000), anti-MDA5 (Cell Signaling, #5321, monoclonal, rabbit, 1:1000), anti-MAVS (Cell Signaling, monoclonal, rabbit, #24930, 1:1000), anti-IRF3 (Cell Signaling, monoclonal, rabbit, #11904, 1:1000), anti-TBK1 (Cell Signaling, polyclonal, rabbit, #3013S, 1:1000), anti-Viperin (Cell Signaling, monoclonal, rabbit, #13996, 1:1000), anti-IFIT1 (Cell Signaling, monoclonal, rabbit, #14769, 1:1000), anti-IFITM3 (Cell Signaling, monoclonal, rabbit, #59212, 1:1000), anti-ISG15 (Cell Signaling, monoclonal, rabbit, #2758, 1:1000), anti-NS1 (Invitrogen, polyclonal, rabbit, #PA5-32243, 1:1000), or anti-STING (Cell Signaling, monoclonal, rabbit, #13647). Anti-Vinculin (Sigma, monoclonal, mouse, #V9131, 1:10000) was used as a loading control. After three washes with PBS/Tween, the membranes were incubated with secondary antibody solutions with peroxidase-conjugated F(ab)2 donkey anti-mouse IgG (Jackson, #711-036-150, 1:10000) or peroxidase-conjugated F(ab)2 donkey anti-rabbit IgG (Jackson, #711-036-152, 1:10000) for 1–2 hours at room temperature. The membranes were images using an Image Quant LAS4000 mini (GE Healthcare), exposing which either SuperSignal West Pico PLUS chemiluminescent substrate (Thermo, # 34577) or the SuperSignal West Femto chemiluminescent substrate (Thermo, #34096). Uncropped images are available in S2 Fig.

## Reverse transcriptase-quantitative polymerase chain reaction (RT-qPCR)

RT-qPCR was used to quantify the transcription of genes of interest. RNA was lysed and isolated from the samples using the High Pure RNA Isolation Kit (Life Science, #11828665001) according to the manufacturer's protocol. RNA concentrations were measured with a NanoDrop spectrophotometer (Thermo Fisher). Gene expression was analyzed using premade TaqMan assays and the RNA-to-Ct-1-Step kit (Applied Biosciences), following manufacturer's recommendations. The RT-qPCR was performed on an AriaMx Real-Time PCR System. TaqMan assays for qPCR were purchased from Applied Bioscience; GAPDH (Hs02786624), CXCL10 (Hs00171042), ISG15 (Hs01921425), FOXJ1 (Hs00230964), SCGB1A1 (Hs00171092), MUC5AC (Hs01365616), MUC5B (Hs06629268) IFNB1 (Hs01077958), and IFNL1 (Hs00601677). Furthermore, we used custom made TaqMan primers for:

NP – segment 5: Forward 5'-GGAAATTTCAAACTGCTGCACAAAA, Reverse
5'- CGTGCTAGAAAAGTGAGATCTTCGA

M2 – segment 7: Forward 5'-AACCTGTGAACAGATTGCTGACT, Reverse 5'-TCTGATTAGTGGGTTGGTTGTTGT

NS1 – segment 8: Forward 5'- GCTAAGGGCTTTCACCGAAGAG, Reverse
5'- TGGAAGAGAAGGCAATGGTGAAATT

## hCXCL-10 ELISA

To quantify the release of CXCL-10 from the HAE-ALI cultures, we used hCXCL-10 ELISA. The ELISAs were performed on apical mucus washes and on basolateral media collected prior to cell harvest. A 96 well, half-area, flat-bottom plate (Corning, 3690, VWR) was coated with a capture antibody mix from the CXCL10 DuoSet ELISA (R&D, #DY266), prepared according to manufacturer's protocol, and incubated ON at room temperature. The plate was washed with a wash buffer consisting of PBS (Biowest, #X0515, mixed with 4,5 L of deionized $H_2O$) with 0.05% Tween 20 (Sigma-Aldrich, P1379). This wash was used three times between each step of the ELISA. The plate was blocked for a minimum of 1 hour using a Reagent Diluent with 1% BSA (Sigma, #9048-46-8) in PBS. The standard dilutions (R&D, #DY266, prepared using manufacturer's protocol), blanks and sample dilutions (undiluted for 16 hours, diluted 1:10 for mucus 48 h p.i. and 1:20 for media 48 h p.i.) was added and incubated ON at 4°C. The following day, a detection antibody mix (R&D, #DY266, prepared using manufacturer's protocol) was added and left to incubate at room temperature for 2 hours. Next, a 1:40 dilution of Streptavidin-HPR B (R&D, #DY266, Part-number 893975) in Reagent Diluent was added for 20 min at RT in the dark. After a final wash, 50 µL of the substrate solution TMB One Solution (Promega, #G7431) was added and incubated in the dark until the color development was sufficient. The reaction was stopped after approx. 10–20 minutes by adding 25 µl of stop-solution (1 mol/L (2N) sulfuric acid ($H_2SO_4$), VWR Chemicals). Results were acquired with a plate reader (BioTek Synergy HTX Multimode Reader) with optical density determined at 450 nm, using 570 nm for wavelength correction.

## Fluorescence staining of HAE-ALI cultures

Before staining, the AAVS1 KO and MAVS KO HAE-ALI membranes were washed for 5 minutes to remove mucus build-up by adding 200 µL of DMEM. The basolateral media was removed and the Transwell inserts were moved into a new 24-well plate (Sarstedt, #83.3922). Cells were fixed using 4% formaldehyde solution (16% formaldehyde solution, Methanol-free, (Pierce #28908), diluted in PBS) by adding 200 µL of the solution to the apical surface and 400 µL to the basolateral compartment for 20–30 minutes at room temperature. After fixation, both compartments were washed three times with PBS. Plates can be stored at 4°C for approx. 1 month if 250 µL PBS with Sodium azide (1:100 dilution) is added to the apical surface and 800 µL to the basolateral compartment. Before staining, the cells were permeabilized by adding freshly made 0.2% Triton-X100 (in PBS) to the apical surface and blocked using 0,5% BSA (in PBS). After blocking, the membranes were removed from the Transwell inserts by cutting around the outer edge with a scalpel. Each membrane was divided into four pieces, that were stained separately. Primary antibody solutions (diluted in the 0,5% BSA blocking buffer) were added for 1–2 hours. After three washes with blocking buffer, the membrane pieces were incubated for 1 hour in the dark with secondary antibodies diluted 1:500 in blocking buffer. For some of the membranes, a DAPI stain was added during the last 5 minutes of the incubation. The staining of the membrane pieces was as follows:

- Primary: Anti-β-tubulin (Abcam, monoclonal, rabbit, #Ab179509, 1:500) and anti-CC10 (Santa Cruz Biotechnology, monoclonal, mouse, #sc-365992, 1:200). Secondary: Anti-rabbit-Alexa Flour 488 (Invitrogen, polyclonal, donkey, #A-32790) and anti-mouse Alexa Fluor 647(Invitrogen, polyclonal, chicken, #A-21463).

- Primary: Anti-MUC5AC (Cell Signaling, monoclonal, rabbit, #61193, 1:200) and anti-CC10 (Santa Cruz Biotechnology, monoclonal, mouse, #sc-365992, 1:200). Secondary: Anti-rabbit-Alexa Flour 488 (Invitrogen, polyclonal, donkey, #A-32790) and anti-mouse Alexa Fluor 647 (Invitrogen, polyclonal, chicken, #A-21463).

- Primary: Anti-HA (Abcam, monoclonal, mouse, #ab8262, 1:500). Secondary: Anti-mouse Alexa Fluor 647 (Invitrogen, polyclonal, chicken, #A-21463) and DAPI (Sigma Aldrich, #D9542, 1:500)

- Primary: Anti-NS1 (Invitrogen, polyclonal, rabbit, #PA5-32243, 1:200). Secondary: Anti-rabbit Alexa Fluor 647 (Invitrogen, polyclonal, goat, #A-21245) and DAPI Sigma Aldrich, #D9542, 1:500).

Before mounting, each membrane piece was washed three times with blocking buffer. One drop of ProLong Glass Anti-fade Mountant (Invitrogen, #P36982) was added to a glass slide. The membrane pieces were blotted against paper tissue and placed in the mounting media cell side up. A coverslip was added on top, and the slide was allowed to dry horizontally ON before imaging. Fluorescence images were acquired using an Olympus BX63 Upright Widefield Fluorescence microscope with a 40x, Plan Flurite objective and a Sensitive Andor Zyla 5.5 camera. Images were acquired and analyzed at the Bioimaging Core Facility, Health, Aarhus University, Denmark. All images were acquired using identical microscope settings. Brightness and contrast were adjusted in ImageJ for visualization.

## Statistical analysis

Graphs and statistics were created and calculated using Graph Pad Prism 10. For all panels, bars represent mean $\pm$ s.e.m. Unless otherwise stated in the figure legend, statistical significance was assessed using multiple unpaired t-tests. The false discovery rate (FDR) in both unpaired and paired multiple t-tests was controlled by the two-stage step-up method of Benjamini, Krieger and Yekutieli with a desired FDR = 5%. Q-values are indicated above each comparison. Results with a $q < 0.05$ were considered significant.

## Quantification of Western Blotting

Membrane images from Western Blotting were imported into the AzureSpot Pro 1.5 software. Membranes blotted for either NS1 or Vinculin were analyzed separately. Lanes were defined by adding a box with the correct number of lanes and adjusting each box manually to include the band. Background was subtracted using "Rolling Ball" with a radius of 40. In all lane, the band was detected manually with a fixed band size of 30 pixels. The volumes of each band were exported to an Excel sheet where the volume of NS1 was normalized to the volume of Vinculin.

## Quantification of immunofluorescence images (IF)

DAPI- and NS1-stained cells were quantified from IF images acquired at 40x magnification. The images were imported into ImageJ2 (version 2.16.0), and macros were created to quantify cells numbers. The following ImageJ tools were applied to all DAPI-stained images: Subtract Background (rolling = 100), Gaussian Blur (sigma = 4 slice), Enhance Contrast (saturated = 0.50 normalize), Auto Threshold (Default, dark), Convert to Mask, Watershed, Fill Holes, and Analyze Particles (size = 10-Infinity). To estimate the number of NS1-positive cells, a second macro was applied: Subtract Background (rolling = 100), Gaussian Blur (sigma = 2 slice), Enhance Contrast (saturated = 0.35 normalize), Auto Threshold (Triangle, dark), Convert to Mask, Close, Fill Holes, Distance Map, Find Maxima (prominence = 10 exclude, output = Maxima Within Tolerance), and Analyze Particles (size = 10-Infinity). For each macro, the "Count" value from the summary of each image was imported to an Excel sheet, and the number of NS1-positive cells was normalized to the total cell count.

## Ethics

Samples were collected from adult healthy human subjects. Ethical approval was provided by the Danish Council on Ethics (1-10-72-182-19). Written informed consent was obtained from all participants witnessed in writing by a third party. Documentation for consent has been archived locally. Recruitment period started on 01/05/2024 and ended 01/08/2025.

## Supporting information

**S1 Fig. Supplementary figures showing additional interferon reponses and viral replication during shorter infections.**
(PDF)

**S2 Fig. Supplementary figures containing uncropped immunoblots.**
(PDF)

## Acknowledgments

We would like to acknowledge the Bioimaging Core Facility, Health, Aarhus University, Denmark, for the use of equipment and support.

## Author contributions

**Conceptualization:** Maja Hemberg, Anne Louise Hansen, Christian Kanstrup Holm.

**Formal analysis:** Maja Hemberg, Christian Kanstrup Holm.

**Funding acquisition:** Christian Kanstrup Holm.

**Investigation:** Maja Hemberg, Anne Louise Hansen, Jacob Storgaard, Julia Blay-Cadanet, Alice Pedersen, Anne Laugaard Thielke, Christian Kanstrup Holm.

**Methodology:** Maja Hemberg, Anne Louise Hansen, Jacob Storgaard, Julia Blay-Cadanet, Alice Pedersen, Anne Laugaard Thielke, Christian Kanstrup Holm.

**Project administration:** Maja Hemberg, Anne Louise Hansen, Julia Blay-Cadanet, Christian Kanstrup Holm.

**Supervision:** Anne Louise Hansen, Christian Kanstrup Holm.

**Writing – original draft:** Maja Hemberg.

**Writing – review & editing:** Maja Hemberg, Christian Kanstrup Holm.

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
