## [Decision Letter · Decision Letter 0]

21 Apr 2026

PONE-D-26-09926MAVS is Important for Antiviral Defense Against Influenza A Virus in a Human Respiratory Epithelium ModelPLOS One

Dear Dr. Holm,

Thank you for submitting your manuscript to PLOS ONE. After careful consideration, we feel that it has merit but does not fully meet PLOS ONE’s publication criteria as it currently stands. Therefore, we invite you to submit a revised version of the manuscript that addresses the points raised during the review process.

We look forward to receiving your revised manuscript.

Kind regards,

Gilberto Jose Betancor Quintana, Ph.D

Academic Editor

PLOS One

Journal Requirements:

“Novo Nordisk Fonden (NNF):Maja Hemberg,Anne Louise Hansen,Jacob Storgaard,Julia Blay-Cadanet,Alice Pedersen,Anne Laugaard Thielke,Christian K Holm 0066798; Frimodt-Heineke Fonden (Frimodt Heineke Fonden)”

“This work was supported by Frimodt-Heineke Fonden, Dagmar Marshalls Fond, the Independent Research Fond Denmark (4287-00003B) and Hallas Møller Ascending Investigator (#0066798). We would like to acknowledge the Bioimaging Core Facility, Health, Aarhus University, Denmark, for the use of equipment and support.”

“Novo Nordisk Fonden (NNF):Maja Hemberg,Anne Louise Hansen,Jacob Storgaard,Julia Blay-Cadanet,Alice Pedersen,Anne Laugaard Thielke,Christian K Holm 0066798; Frimodt-Heineke Fonden (Frimodt Heineke Fonden)”

Additional Editor Comments:

Dear authors,

After careful consideration and based on the opinion of previous and new reviewers, we believe that your article "MAVS is Important for Antiviral Defense Against Influenza A Virus in a Human Respiratory Epithelium Model" is of enough quality and impact for publication in PLoS One, pending on completion of some modifications.

As you can see on the reviewers comments, they believe further discussion on the limitations of the virus strains employed, the method use to quantify viral replication and extend of gene knock out are needed. In addition, some text corrections are proposed.

Kind regards,

Reviewer's Responses to Questions

**Comments to the Author**

1. Is the manuscript technically sound, and do the data support the conclusions?

Reviewer #1: Yes

Reviewer #2: Yes

2. Has the statistical analysis been performed appropriately and rigorously? 

Reviewer #1: Yes

Reviewer #2: Yes

3. Have the authors made all data underlying the findings in their manuscript fully available?

Reviewer #1: Yes

Reviewer #2: Yes

4. Is the manuscript presented in an intelligible fashion and written in standard English?

Reviewer #1: Yes

Reviewer #2: Yes

5. Review Comments to the Author

Reviewer #1: Interferon-based antiviral immunity is clearly of fundamental importance and has been extensively demonstrated in humans, mouse models, and cultured cell line systems. However, much less is known about how interferon-mediated immunity functions in the nasal mucosal epithelium, which serves as the initial physical barrier encountered by inhaled viruses. In mouse models, interferon-producing immune cells often do not accumulate at sites of infection immediately during the earliest phase of infection, highlighting the importance of understanding the intrinsic antiviral capacity of the nasal epithelium itself. In this study, the authors used electroporation to generate MAVS-ko nasal epithelial cultures and demonstrated the critical role of MAVS in interferon responses and antiviral defense. Overall, the methodology established here will provide an important foundation for future studies in this field.

Major comments

1. The use of the IAV PR8 strain somewhat limits the relevance of this study to influenza virus infection in its more physiological context. However, the inclusion of SeV partially addresses this limitation and suggests that the observed effect is not restricted to influenza virus alone. This point would be worth discussing more explicitly in the Discussion.

2. Genetic manipulation in primary human nasal epithelial cells is more challenging than is often appreciated. Electroporation can compromise stemness and differentiation capacity to some extent. We therefore appreciate the authors’ transparent presentation of the finding that MAVS knockout was also associated with reduced MUC5AC expression. However, under these conditions, it is difficult to determine whether this reduction reflects impaired differentiation or a direct role of MAVS in epithelial development. It is unfortunate that cilia staining was not performed in the MAVS-knockout nasal epithelial cultures, as this would have helped assess whether epithelial differentiation remained intact.

3. As the authors note in the Discussion, it is unfortunate that the immunofluorescence analysis did not include co-staining with cilia markers and MUC5AC. Such experiments would likely have provided more informative insight into the epithelial phenotype.

4. I suggest expanding the Discussion to emphasize the following point: mouse studies have firmly established the importance of interferon responses in antiviral defense. However, in vivo, the interplay among nasal epithelial cells, innate immune cells, and adaptive immune cells makes it difficult to define the tissue-specific and time-dependent contributions of interferon-mediated antiviral activity. In this context, the current experimental system provides a particularly direct approach to examine the intrinsic antiviral capacity of the nasal epithelium, the first respiratory tissue encountered by inhaled viruses, during the first 24 to 48 hours of infection before substantial immune cell recruitment occurs.

Minor comment

Because CC10/SCUB1A1 is used here as a club cell marker, it would be helpful to define this explicitly in the main text rather than only in the figure legend.

Reviewer #2: This study provides a compelling investigation into the role of MAVS in the antiviral response to influenza A, distinguished by its use of a highly physiological HAE-ALI model. The methodological integration of CRISPR-Cas9-mediated gene disruption in primary human cells is a significant strength, and the data generally support the authors' conclusions.

The manuscript is technically sound and well-written. To maximize the impact of the work, the authors should provide further clarity on the limitations of the viral replication assays, address the uniformity of gene editing, and justify the selection of virus strains. Since many of my concerns (outlined below) have been clearly acknowledged as limitations in the Discussion, I recommend accepting the manuscript for publication after minor modification.

Major Comments

1- The primary novelty of the study lies in the use of a primary human respiratory epithelium model to interrogate MAVS function, rather than in uncovering a fundamentally new signaling mechanism. This should be emphasized clearly in the Introduction and Discussion to appropriately frame the contribution. While the conclusion that MAVS is non-redundant is supported within this model, caution should be taken when extrapolating this conclusion beyond epithelial-only systems.

2- Increased viral gene expression and NS1 protein abundance suggest enhanced viral replication in MAVS KO cultures. However, the absence of direct quantification of infectious viral titers (e.g., plaque assay or TCID50) limits the strength of this conclusion.

3- Residual MAVS expression is detectable by Western blot, suggesting incomplete knockout. Given the heterogeneity of HAE-ALI cultures, the lack of cell-type–specific assessment of knockout efficiency (e.g., ciliated vs. secretory cells) should be acknowledged as a limitation, particularly because influenza virus displays cell-type–dependent tropism.

4- The use of the laboratory-adapted A/PR/8/34 strain is reasonable for mechanistic studies, but it may not fully capture host–pathogen interactions relevant to contemporary human influenza viruses. The Discussion would benefit from explicitly stating this limitation and suggesting that future studies validate findings using clinical isolates.

5- The study is limited to two donors of the same sex. While understandable given the complexity of the model, this should be stated explicitly as a limitation.

6- Supplementary Figure F should be “…for 16 h”. In addition, it is unclear what distinguishes Fig. 2A from Supplementary Figure A, as well as Fig. 2C from Supplementary Figure B. Please clarify these differences explicitly in the figure legends.

6. PLOS authors have the option to publish the peer review history of their article (what does this mean?). If published, this will include your full peer review and any attached files.

Reviewer #1: **Yes:** Chien-Ting Wu

Reviewer #2: **Yes:** Wenxin Wu

---

## [Author Response · Author response to Decision Letter 1]

18 May 2026

Response to Reviewers

Reviewer #1: Interferon-based antiviral immunity is clearly of fundamental importance and has been extensively demonstrated in humans, mouse models, and cultured cell line systems. However, much less is known about how interferon-mediated immunity functions in the nasal mucosal epithelium, which serves as the initial physical barrier encountered by inhaled viruses. In mouse models, interferon-producing immune cells often do not accumulate at sites of infection immediately during the earliest phase of infection, highlighting the importance of understanding the intrinsic antiviral capacity of the nasal epithelium itself. In this study, the authors used electroporation to generate MAVS-ko nasal epithelial cultures and demonstrated the critical role of MAVS in interferon responses and antiviral defense. Overall, the methodology established here will provide an important foundation for future studies in this field.

We would like to thank Reviewer 1 for their assessment of our manuscript. We are pleased with the overall positive comments, especially that the reviewer appreciates the potential of the methodology used in our study. However, this reviewer also raised some concerns, that we will address below:

Major comments:

1. The use of the IAV PR8 strain somewhat limits the relevance of this study to influenza virus infection in its more physiological context. However, the inclusion of SeV partially addresses this limitation and suggests that the observed effect is not restricted to influenza virus alone. This point would be worth discussing more explicitly in the Discussion.

This is a very valid point. We have addressed this concern by adjusting the discussion of this limitation to the following:

“Notably, inclusion of SeV, a robust activator of RIG-I signaling, supports that the observed MAVS-dependent effects are not restricted to IAV but reflect a more general role in antiviral protection of the respiratory epithelium. Despite these limitations, the use of primary human respiratory epithelium cells remains a major strength of this study, providing a physiologically relevant system to investigate antiviral signaling.” (Page 14, line nr. 307-312)

2. Genetic manipulation in primary human nasal epithelial cells is more challenging than is often appreciated. Electroporation can compromise stemness and differentiation capacity to some extent. We therefore appreciate the authors’ transparent presentation of the finding that MAVS knockout was also associated with reduced MUC5AC expression. However, under these conditions, it is difficult to determine whether this reduction reflects impaired differentiation or a direct role of MAVS in epithelial development. It is unfortunate that cilia staining was not performed in the MAVS-knockout nasal epithelial cultures, as this would have helped assess whether epithelial differentiation remained intact.

We agree that electroporation of the primary cells may affect epithelial differentiation, which could confound our findings. We hope figure 1I addresses this concern, since ciliated cell staining was performed using-tubulin (cellular marker for ciliated cells) on both AAVS1 and MAVS KO cultures. Based on this analysis, we did not observe differences in the presence or pattern of ciliated cells between the two KO cultures. In addition, qPCR analysis of the ciliated cell marker FOXJ1 (fig. 1H) did not indicate a difference in ciliated cell-transcripts. Together, these data suggest that major defects in epithelial differentiation are unlikely between the two cultures. However, we cannot fully exclude more subtle effects on epithelial cell differentiation or composition, and this has now been acknowledged in the discussion.

“Nevertheless, analysis of FOXJ1, SCGB1A1, MUC5AC, and MUC5B expression, together with-tubulin staining, indicated no overall changes in epithelial composition. However, we cannot exclude subtle effects of gene editing on epithelial differentiation or composition, which might influence specific cell populations.” (Page 13, line nr. 292 until page 14, line nr. 295)

3. As the authors note in the Discussion, it is unfortunate that the immunofluorescence analysis did not include co-staining with cilia markers and MUC5AC. Such experiments would likely have provided more informative insight into the epithelial phenotype.

We agree that co-staining with cilia markers and MUC5AC would provide additional insight into epithelial cell composition and phenotype. These experiments were not included in the current study, as our immunofluorescence panel did not allow this combination. We agree that such co-staining would be valuable in future studies to further characterize epithelial differentiation and cell-type–specific responses.

4. I suggest expanding the Discussion to emphasize the following point: mouse studies have firmly established the importance of interferon responses in antiviral defense. However, in vivo, the interplay among nasal epithelial cells, innate immune cells, and adaptive immune cells makes it difficult to define the tissue-specific and time-dependent contributions of interferon-mediated antiviral activity. In this context, the current experimental system provides a particularly direct approach to examine the intrinsic antiviral capacity of the nasal epithelium, the first respiratory tissue encountered by inhaled viruses, during the first 24 to 48 hours of infection before substantial immune cell recruitment occurs.

We thank the reviewer for this suggestion, that will help strengthen the discussion of our results. We have added the following paragraph to the discussion:

“In vivo, antiviral immunity arises from complex interactions between epithelial, innate immune, and adaptive immune cells, making it difficult to isolate tissue-specific contributions. The HAE-ALI model used in this study enables focused investigation of epithelial-intrinsic antiviral responses during the early phase of infection, prior to substantial immune cell recruitment. Therefore, this model provides unique insight into the first line of antiviral defense at the respiratory barrier.” (Page 14, line nr. 313-317)

Minor comment

Because CC10/SCUB1A1 is used here as a club cell marker, it would be helpful to define this explicitly in the main text rather than only in the figure legend.

This is a very valid point. CC10/SCGB1A1 has now been defined as a club cell marker in the main text of the paper.

“HAE-ALI cultures from two independent donors (donor 1 and donor 2) were assessed by qPCR for expression of Forkhead Box J1 (FOXJ1), Secretoglobin Family 1A Member 1 (SCGB1A1, also known as CC10), Mucin 5AC (MUC5AC) and Mucin 5B (MUC5B). FOXJ1 was used as a marker for ciliated cells, SCGB1A1 (CC10) as a marker of club cells, and MUC5AC and MUC5B as markers of goblet cells.” (Page 5, line nr. 88-92)

Reviewer #2: This study provides a compelling investigation into the role of MAVS in the antiviral response to influenza A, distinguished by its use of a highly physiological HAE-ALI model. The methodological integration of CRISPR-Cas9-mediated gene disruption in primary human cells is a significant strength, and the data generally support the authors' conclusions.

The manuscript is technically sound and well-written. To maximize the impact of the work, the authors should provide further clarity on the limitations of the viral replication assays, address the uniformity of gene editing, and justify the selection of virus strains. Since many of my concerns (outlined below) have been clearly acknowledged as limitations in the Discussion, I recommend accepting the manuscript for publication after minor modification.

We sincerely thank reviewer #2 for their thoughtful and constructive evaluation of our manuscript. We are grateful for the recognition of the study’s strengths, particularly the use of the physiologically relevant HAE-ALI model and the application of CRISPR-Cas9–mediated gene knockout in primary human cells. We are also pleased that the reviewer found the manuscript to be technically sound and that the data support our conclusions. We thank the reviewer for their suggestions to further strengthen the study, and these comments will be addressed below:

Major Comments

1- The primary novelty of the study lies in the use of a primary human respiratory epithelium model to interrogate MAVS function, rather than in uncovering a fundamentally new signaling mechanism. This should be emphasized clearly in the Introduction and Discussion to appropriately frame the contribution. While the conclusion that MAVS is non-redundant is supported within this model, caution should be taken when extrapolating this conclusion beyond epithelial-only systems.

We thank the reviewer for this important point. We agree that the primary novelty of this study lies in the use of a physiologically relevant primary human epithelial model to investigate MAVS function. We have revised the Introduction and Discussion to more clearly emphasize this aspect. In addition, we have clarified in the Discussion that conclusions regarding the non-redundant role of MAVS are limited to the epithelial system used here and may not fully extend to more complex in vivo settings.

“In particular, the function of MAVS within the respiratory epithelium, the first line of defense against IAV, has not been directly investigated in a physiologically relevant human model. Understanding epithelial-intrinsic MAVS signaling is important for determining how early innate immune responses contribute the outcome of IAV infection.” (Page 3, line nr. 51-54)

“The findings build on prior studies identifying MAVS as a central mediator of antiviral signaling; however, its specific role in the respiratory epithelium has not previously been investigated. Using a primary human airway epithelial (HAE-ALI) model, we provide direct evidence of epithelial-intrinsic antiviral responses regulated by MAVS. We show that MAVS is required for the induction IFNs and ISGs following IAV infection, and that viral replication is increased during prolonged infection in its absence in this HAE-ALI model.” (Page 12, line nr. 249-254)

“However, these findings should be interpreted within an epithelial-only system, as complex interactions between epithelial and immune cells in vivo contribute to antiviral defenses. Therefore, the extent to which MAVS-dependent signaling is compensated by other pathways in more complex systems remains to be determined. Overall, MAVS KO significantly impairs IFN and ISG induction in this physiologically relevant human respiratory epithelial model, highlighting a critical role for MAVS in respiratory epithelial antiviral defense.” (Page 15, line nr. 323-328)

2- Increased viral gene expression and NS1 protein abundance suggest enhanced viral replication in MAVS KO cultures. However, the absence of direct quantification of infectious viral titers (e.g., plaque assay or TCID50) limits the strength of this conclusion.

We agree with this comment and have clarified this limitation in the discussion further:

“Although increased viral transcripts and altered viral protein levels in MAVS KO cultures indicate enhanced viral burden, infectious viral titers were not directly quantified by plaque-forming or TCID-50 assays, which remain the gold standards for assessing replication kinetics. Therefore, while our data support increased viral activity in the absence of MAVS, the extent to which this reflects an increased production of infectious virions cannot be definitively determined and needs further investigation.” (Page 13, line nr. 270-275)

3- Residual MAVS expression is detectable by Western blot, suggesting incomplete knockout. Given the heterogeneity of HAE-ALI cultures, the lack of cell-type–specific assessment of knockout efficiency (e.g., ciliated vs. secretory cells) should be acknowledged as a limitation, particularly because influenza virus displays cell-type–dependent tropism.

This is a very valid point that has already been partially presented in the discussion. However, this section has been rewritten to enhance the point that incomplete KO and cell-type heterogeneity could specifically influence interpretation of viral replication and IFN responses:

“Although MAVS protein levels were markedly reduced, low residual expression remained detectable by western blot. HAE-ALI cultures are heterogeneous, and electroporation may not affect all cells, making complete MAVS elimination challenging. Residual IFNs and ISGs induction in MAVS KO cultures may reflect incomplete gene disruption in a subset of cells and/or compensatory signaling through alternative pathways.

Although both ciliated and secretory epithelial populations were confirmed in KO and control cultures, gene editing efficiency was not assessed at single-cell resolution. Given that IAV exhibits strain- and cell-type-dependent tropism, heterogeneous editing across epithelial subtypes may influence observed antiviral responses. However, MAVS KO cultures from both donors consistently showed reduced protein expression compared to controls, supporting effective pathway disruption at population level and allowing investigation of the role of MAVS in this respiratory epithelium model.” (Page 13, line nr. 277-288)

4- The use of the laboratory-adapted A/PR/8/34 strain is reasonable for mechanistic studies, but it may not fully capture host–pathogen interactions relevant to contemporary human influenza viruses. The Discussion would benefit from explicitly stating this limitation and suggesting that future studies validate findings using clinical isolates.

We agree that the use of the laboratory-adapted influenza A/PR/8/34 strain represents a limitation. This point has already been more explicitly stated in the discussion:

“Consequently, the host-pathogen interactions observed in this study may differ from those elicited by contemporary clinical influenza isolates. This represents a limitation of the present study, and future work using circulating human influenza strains will be important to determine broader applicability.” (Page 14, line nr. 304-307)

5- The study is limited to two donors of the same sex. While understandable given the complexity of the model, this should be stated explicitly as a limitation.

We thank the reviewer for this comment. We agree that the use of only two same-sex donors represents a limitation of the present study. This has now been more explicitly stated in the discussion.

“This study is limited by the use of two same-sex donors. Although responses were consistent across both donors, inclusion of additional donors of both sexes would improve generalizability. In addition, the HAE-ALI model lacks immune cell components, and therefore does not capture epithelial-immune interactions that contribute to antiviral defense in vivo. As such, our findings reflect epithelial-intrinsic MAVS function rather than whole-tissue immunity.” (Page 14, line nr. 297-301)

6- Supplementary Figure F should be “…for 16 h”. In addition, it is unclear what distinguishes Fig. 2A from Supplementary Figure A, as well as Fig. 2C from Supplementary Figure B. Please clarify these differences explicitly in the figure legends.

We thank the reviewer for bringing this to our attention. The mistake has been corrected, so the figure legend for supplementary figures F now reads:

Suppl. fig. F: “RT-qPCR analysis of viral RNA from three different viral segments from AAVS1 or MAVS KO HAE-ALI cultures infected with IAV (MOI 0.5) for 16 hours. Statistical difference was determined by an unpaired Welch’s t-test; “ns” indicates p-value > 0,05.”

Distinguishing figures in figure 2 from the similar figures in supplementary figures, is another important point by the reviewer. The figure legends have now been changes to specify which donor the result in each figure is referring to:

Fig. 2A: “RT-qPCR analysis of IFNB1 and IFNL1 in AAVS1 or MAVS KO HAE-ALI cultures infected with IAV (MOI 0.5) or left uninfected for 16 hours. Each data point represents an independent culture (n=3) derived from the same donor (donor 2). Data is representative of two independent experiments (see Suppl. Fig. A). For IFNL1, some uninfected cultures had undetectable

---

## [Editor Report · Decision Letter 1]

19 May 2026

MAVS is Important for Antiviral Defense Against Influenza A Virus in a Human Respiratory Epithelium Model

PONE-D-26-09926R1

Dear Dr. Holm,

We’re pleased to inform you that your manuscript has been judged scientifically suitable for publication and will be formally accepted for publication once it meets all outstanding technical requirements.

Kind regards,

Gilberto Jose Betancor Quintana, Ph.D

Academic Editor

PLOS One

Additional Editor Comments (optional):

Dear authors,

Thank you for submitting the latest version of your manuscript,

I have revised it and I believe you have satisfactory answered to all the reviewers comments. Therefore, I have accepted the article for publication,

Kind regards,

The academic editor
---

## [Editor Report · Acceptance letter]

PONE-D-26-09926R1

PLOS One

Dear Dr. Holm,

I'm pleased to inform you that your manuscript has been deemed suitable for publication in PLOS One. Congratulations! Your manuscript is now being handed over to our production team.

Kind regards,

on behalf of

Dr. Gilberto Jose Betancor Quintana

Academic Editor

PLOS One